# The Dantu blood group prevents parasite growth *in vivo*: Evidence from a controlled human malaria infection study

Silvia N Kariuki[1]\*, Alexander W Macharia[1], Johnstone Makale[1], Wilfred Nyamu[1], Stephen L Hoffman[2], Melissa C Kapulu[1,3], Philip Bejon[1,3], Julian C Rayner[4], Thomas N Williams[1,5], On behalf of for the CHMI-SIKA Study Team

[1]Centre for Geographic Medicine Research (Coast), KEMRI-Wellcome Trust Research Programme, Kilifi, Kenya; [2]Sanaria Incorporate, Rockville, United States; [3]Centre for Tropical Medicine and Global Health, Nuffield Department of Medicine, University of Oxford, Oxford, United Kingdom; [4]Cambridge Institute for Medical Research, University of Cambridge, Cambridge, United Kingdom; [5]Institute for Global Health Innovation, Department of Surgery and Cancer, Imperial College London, London, United Kingdom

\*For correspondence:
SNKariuki@kemri-wellcome.org

## Abstract

**Background:** The long co-evolution of *Homo sapiens* and *Plasmodium falciparum* has resulted in the selection of numerous human genetic variants that confer an advantage against severe malaria and death. One such variant is the Dantu blood group antigen, which is associated with 74% protection against severe and complicated *P. falciparum* malaria infections in homozygous individuals, similar to that provided by the sickle haemoglobin allele (HbS). Recent *in vitro* studies suggest that Dantu exerts this protection by increasing the surface tension of red blood cells, thereby impeding the ability of *P. falciparum* merozoites to invade them and reducing parasite multiplication. However, no studies have yet explored this hypothesis *in vivo*.

**Methods:** We investigated the effect of Dantu on early phase *P. falciparum* (Pf) infections in a controlled human malaria infection (CHMI) study. 141 sickle-negative Kenyan adults were inoculated with $3.2 \times 10^3$ aseptic, purified, cryopreserved Pf sporozoites (PfSPZ Challenge) then monitored for blood-stage parasitaemia for 21 days by quantitative polymerase chain reaction (qPCR)analysis of the 18S ribosomal RNA *P. falciparum* gene. The primary endpoint was blood-stage *P. falciparum* parasitaemia of ≥500/μl while the secondary endpoint was the receipt of antimalarial treatment in the presence of parasitaemia of any density. On study completion, all participants were genotyped both for Dantu and for four other polymorphisms that are associated with protection against severe falciparum malaria: $\alpha^+$-thalassaemia, blood group O, G6PD deficiency, and the rs4951074 allele in the red cell calcium transporter *ATP2B4*.

**Results:** The primary endpoint was reached in 25/111 (22.5%) non-Dantu subjects in comparison to 0/27 (0%) Dantu heterozygotes and 0/3 (0.0%) Dantu homozygotes (p=0.01). Similarly, 49/111 (44.1%) non-Dantu subjects reached the secondary endpoint in comparison to only 7/27 (25.9%) and 0/3 (0.0%) Dantu heterozygotes and homozygotes, respectively (p=0.021). No significant impacts on either outcome were seen for any of the other genetic variants under study.

**Conclusions:** This study reveals, for the first time, that the Dantu blood group is associated with high-level protection against early, non-clinical, *P. falciparum* malaria infections *in vivo*. Learning more about the mechanisms involved could potentially lead to new approaches to the prevention or treatment of the disease. Our study illustrates the power of CHMI with PfSPZ Challenge for directly testing the protective impact of genotypes previously identified using other methods.

**Funding:** The Kenya CHMI study was supported by an award from Wellcome (grant number 107499). SK was supported by a Training Fellowship (216444/Z/19/Z), TNW by a Senior Research Fellowship (202800/Z/16/Z), JCR by an Investigator Award (220266/Z/20/Z), and core support to the KEMRI-Wellcome Trust Research Programme in Kilifi, Kenya (203077), all from Wellcome. The funders had no role in study design, data collection and interpretation, or the decision to submit the work for publication. For the purpose of Open Access, the authors have applied a CC BY public copyright license to any Author Accepted Manuscript version arising from this submission.
**Clinical trial number:** NCT02739763

## Editor's evaluation

The large genetic association studies conducted in East Africa have shown that the Dantu blood group provides substantial protection against severe malaria since it increases the surface tension of red blood cells making it harder for malaria parasites to invade. In this important work, the authors show that parasite growth is indeed restricted *in vivo* by testing this hypothesis in adult Kenyan volunteers infected with *P. falciparium* under careful monitoring. They were able to show convincingly that indeed, parasite growth was reduced amongst Dantu adults.

## Introduction

*Plasmodium falciparum* malaria has been the pre-eminent cause of child morbidity and mortality in the tropics and sub-tropics for much of the last 5000 y. As a consequence, it has had a substantial impact on the human genome through the positive selection of multiple polymorphisms that confer a survival advantage against the disease (*Williams, 2017*). The best studied affect the biology of red blood cells (RBCs), which host malaria parasites for most of their life cycle in humans, the rs334 A>T β$^s$ sickle mutation in *HBB* (*Williams, 2016*), α-thalassaemia (*Williams et al., 2005*), and blood group O (*Fry et al., 2008*) all being important examples.

Recently, we identified a new variant which is associated with high-level protection against severe *P. falciparum* malaria to a degree that is close to that of sickle cell trait (HbAS), the strongest malaria-protective condition yet described (*Malaria Genomic Epidemiology Network, 2014*). The rare Dantu blood group antigen, which results from a genetic rearrangement within the glycophorin (*GYP*) cluster, was shown to confer 74% protection against severe malaria in homozygous individuals (*Band et al., 2015*; *Ndila et al., 2018*). Subsequent *in vitro* studies have suggested that this protection is explained by the resistance of Dantu RBCs to invasion by *P. falciparum* merozoites (*Kariuki et al., 2020*), thereby preventing infections from progressing to become severe or ultimately fatal. However, this hypothesis has not been tested directly *in vivo* to date.

In this study, we have investigated the impact of the Dantu blood group on *in vivo P. falciparum* parasite growth and clinical disease progression through a controlled human malaria infection (CHMI) study with aseptic, purified, cryopreserved *P. falciparum* sporozoites (PfSPZ Challenge) conducted in semi-immune Kenyan adults. To the best of our knowledge, this is the first time that CHMI has been used to directly explore the impact of Dantu on parasite growth *in vivo*.

## Methods

### Study design and population

The primary aim of the Kenya CHMI study was to investigate the impacts of naturally acquired immunity on early-phase malaria infections (*Kapulu et al., 2018*; *Kapulu et al., 2021*). Briefly, 161 healthy adult volunteers living in areas of varying malaria transmission were inoculated by direct venous inoculation (DVI) with $3.2 \times 10^3$ *P. falciparum* sporozoites (PfSPZ) of Sanaria PfSPZ Challenge (NF54) (*Roestenberg et al., 2013*; *Mordmüller et al., 2015*). With a sample size of 161 individuals, we had 80% power to detect a single variable with an effect size ($r^2$) of 0.3 that accounts for 15% of the variability in parasite growth, as previously described (*Kapulu et al., 2018*; *Kapulu et al., 2021*). Because of its major impacts on both malaria susceptibility (*Allison, 1954*) and disease progression (*Taylor et al., 2012*), and in view of results from a previous CHMI with PfSPZ Challenge conducted in Gabon

(*Lell et al., 2018*), recruitment was restricted to those who were negative for both sickle cell trait and disease. After inoculation, venous blood samples were collected twice daily from days 7–14, and then once every day from day 15 until the end of the experiment on day 21, and screened for parasitaemia by quantitative polymerase chain reaction (qPCR) analysis of the *P. falciparum* 18S ribosomal RNA gene. For each participant, the endpoint was considered met, and anti-malarial treatment administered, when a threshold of 500 *P. falciparum* parasites/µl was reached. Participants were treated earlier if signs and symptoms were observed in association with blood film positivity at any parasite density, and on day 21 post-inoculation regardless of outcome. While qPCR was used as the primary measurement of parasitaemia because it is much more sensitive than microscopy at low density (*Bejon et al., 2006*), thick film blood smears were also performed as an additional precaution, and participants were treated if they became blood film positive at any density, an approach that accords with that used in other CHMI studies (*Sheehy et al., 2012*; *Murphy et al., 2012*; *Hodgson et al., 2014*; *Kamau et al., 2014*; *Hodgson et al., 2015*; *Seilie et al., 2019*; *Salkeld et al., 2022*). Participants were recruited during 2016, 2017, and 2018 into three successive cohorts from three different malaria transmission zones: Kilifi North (no- to low-transmission) and Kilifi South (moderate transmission), both on the coast, and Ahero (moderate to high transmission) in Western Kenya. We tested for anti-malarial drugs as described in a previous publication (*Kapulu et al., 2021*). Briefly, antimalarial drugs were measured retrospectively in all volunteers using samples collected on both the day before the challenge and 8 days after challenge. Plasma samples were tested by liquid chromatography-tandem mass spectrometry in two independent laboratories. Sulfadoxine, pyrimethamine, and chloroquine levels were measured at the Strathmore University in Nairobi, Kenya, while artemether and dihydroartemisinin concentrations were measured at the Mahidol Oxford Tropical Medicine Research Unit in Bangkok, Thailand. The study was conducted at the KEMRI-Wellcome Trust Research Programme in Kilifi, Kenya, and was registered on ClinicalTrials.gov (NCT02739763).

## Genotyping for Dantu and other malaria-protective variants

Whole blood samples were collected at the point of recruitment into tubes containing EDTA and stored at –80°C pending batch processing at the end of the study. Genomic DNA was extracted following the manufacturer's instructions using a QIAmp 96 DNA QIAcube HT kit on a QIAcube HT System (QIAGEN, Manchester, UK). Genotyping for Dantu was performed using ABI TaqMan SNP genotyping Assays-by-Design primers and probes on an ABI 7900HT PCR machine, as previously described (*Kariuki et al., 2009*). Dantu genotypes were inferred from the rs186873296 *FREM3* allele, which is in strong linkage disequilibrium with the Dantu structural rearrangement (*Band et al., 2015*). The following primer sequence was used for the rs186873296 SNP: ATGTGAAGAAGCTGGGAACC CTGTC[A/G]TACAAGAAATGACAAAGAAAGCTT, with A being the reference allele. For comparative purposes, we also genotyped participants for a range of other polymorphisms that have been reproducibly associated with protection from severe and complicated malaria in other studies. We typed for the common African form of G6PD deficiency, caused by the *G6PD* c.202T mutation (*Clarke et al., 2017*), the blood group O mutation in *ABO* (*Rowe et al., 2007*) and the rs4951074 allele in *ATP2B4* (*Malaria Genomic Epidemiology Network, 2014*) using TaqMan SNP genotyping assays, and for the -α$^{3.7I}$ deletional form of α+-thalassaemia by gap PCR (*Wambua et al., 2006*). To aid ease of comparison across all variants tested, the genotype groups are categorised as 'Homozygous reference' for individuals with two copies of the reference allele, 'Heterozygous' for individuals with one copy of the reference allele and one copy of the derived allele, and 'Homozygous derived' for individuals with two copies of the derived allele.

## Statistical analysis

We considered three distinct outcomes, each capturing a different aspect of the parasitological and clinical progress of malaria infections: (a) whether infections progressed to reach the pre-defined treatment threshold of 500 parasites/µl; (b) whether or not participants received malaria treatment for either clinical or parasitological reasons; (c) the time from inoculation to treatment in participants who did receive treatment. We made (a) the primary endpoint for this analysis because the hypothesis we were testing concerned *in vivo* parasite growth rather than susceptibility to symptoms. We conducted between-genotype comparisons both by univariate analysis and by multivariate analysis with adjustment for other malaria-protective genes, anti-schizont antibody concentration, and

location of residence, of which the latter two were significantly associated with the same outcomes in an earlier analysis of the same cohort (*Kapulu et al., 2022*). We used the Fisher's exact test using the fmsb package (version 0.7.3) in R to test for any differences in the proportions of individuals that reached the pre-defined treatment threshold of 500 parasites/µl and to test for any differences in the proportions of individuals that required treatment across genotype groups. We also compared the treatment outcomes for the heterozygous- and homozygous-derived genotypes, individually, to the homozygous reference genotype by pairwise analysis.

To test for associations between each variant genotype and the treated or untreated categorical outcome, we used multivariate logistic regression using the glm function in the stats package (version 3.6.2) in R using additive models for each variant, where each variant genotype was coded as zero, one, or two copies of the derived allele. Each multivariable model adjusted for other malaria-protective variants, anti-schizont antibody concentration, and location of residence.

We used the Kruskal–Wallis test (stats package version 3.6.2) and the Dunn's test (FSA package version 0.9.3) to investigate between-group differences in maximum parasitaemia. Finally, we compared time to treatment using Kaplan–Meier survival curves with univariate comparisons across genotype groups performed using the Log-Rank test, and Cox regression models for multivariate analyses, using the survival package (version 3.2.13) in R. All statistical analyses were performed using R V3.6.2 (*R Development Core Team, 2017*).

## Results

Of 161 volunteers recruited to the study, 19 were excluded, either because non-CHMI parasite strains were detected in post-CHMI samples (suggesting the presence of coincidental natural infections) (n = 7), or because antimalarial drugs were detected in pre-CHMI samples (n = 12) (*Figure 1*) (*Kapulu et al., 2021*). After these exclusions, data from 142 individuals contributed to the current analysis. Genotyping revealed that 30 of these individuals were either heterozygous or homozygous for the Dantu allele. We did not get Dantu genotype data on one individual due to poor quality of the DNA sample; therefore, Dantu genotype data on 141 out of the 142 participants was used in the downstream analysis (*Figure 1*).

### The Dantu variant protects against *P. falciparum* growth *in vivo*

While infections progressed to the point of reaching the pre-determined treatment threshold of 500 parasites/µl in 25/142 (17.6%) of all volunteers, the proportions varied markedly by Dantu genotype. None of the thirty (0.0%) Dantu carriers reached this threshold in comparison to 25/111 (22.5%) of the non-Dantu individuals (p=0.01) (*Figure 2*, *Table 1*). The difference in the proportion of Dantu heterozygous (0/27; 0%) and non-Dantu individuals reaching the treatment threshold was strongly significant (p=0.004) but did not reach statistical significance in the case of Dantu homozygotes 0/3 (0.0%) because of the small number of individuals in this group.

### Fewer Dantu carriers required malaria treatment

Although a threshold of 500 parasites/µl was considered the primary endpoint of the study and triggered the administration of antimalarial treatment per-protocol, some participants developed symptoms and were therefore treated before reaching this parasitaemia threshold. Only one quarter (7/27; 25.9%) of the Dantu heterozygotes and none (0/3; 0.0%) of the Dantu homozygotes (*Figure 3*, *Table 2*) received antimalarial treatment in comparison to 49/111 (44.1%) of the non-Dantu individuals. While this did not reach statistical significance on univariate analysis, we carried out multivariate regression analysis with the treated or untreated status as the dependent variable, and the Dantu variant as well as other genetic variants (as described below), anti-schizont antibody and location of residence as the independent variables. This analysis revealed that Dantu-carrying subjects overall were administered antimalarial treatment 83% less frequently (OR 0.17; 95% CI 0.04–0.55; p=0.007) than non-Dantu individuals (*Table 3*). In order to address the core issue of whether prior immunity was a confounder in our analysis, we used measurements of antibodies to whole schizont extract as a proxy indicator of transmission setting or 'malaria exposure' in our multivariate analyses. We compared anti-schizont antibody levels across Dantu genotype groups and found no differences (p=0.659) (*Figure 3—figure supplement 1*).

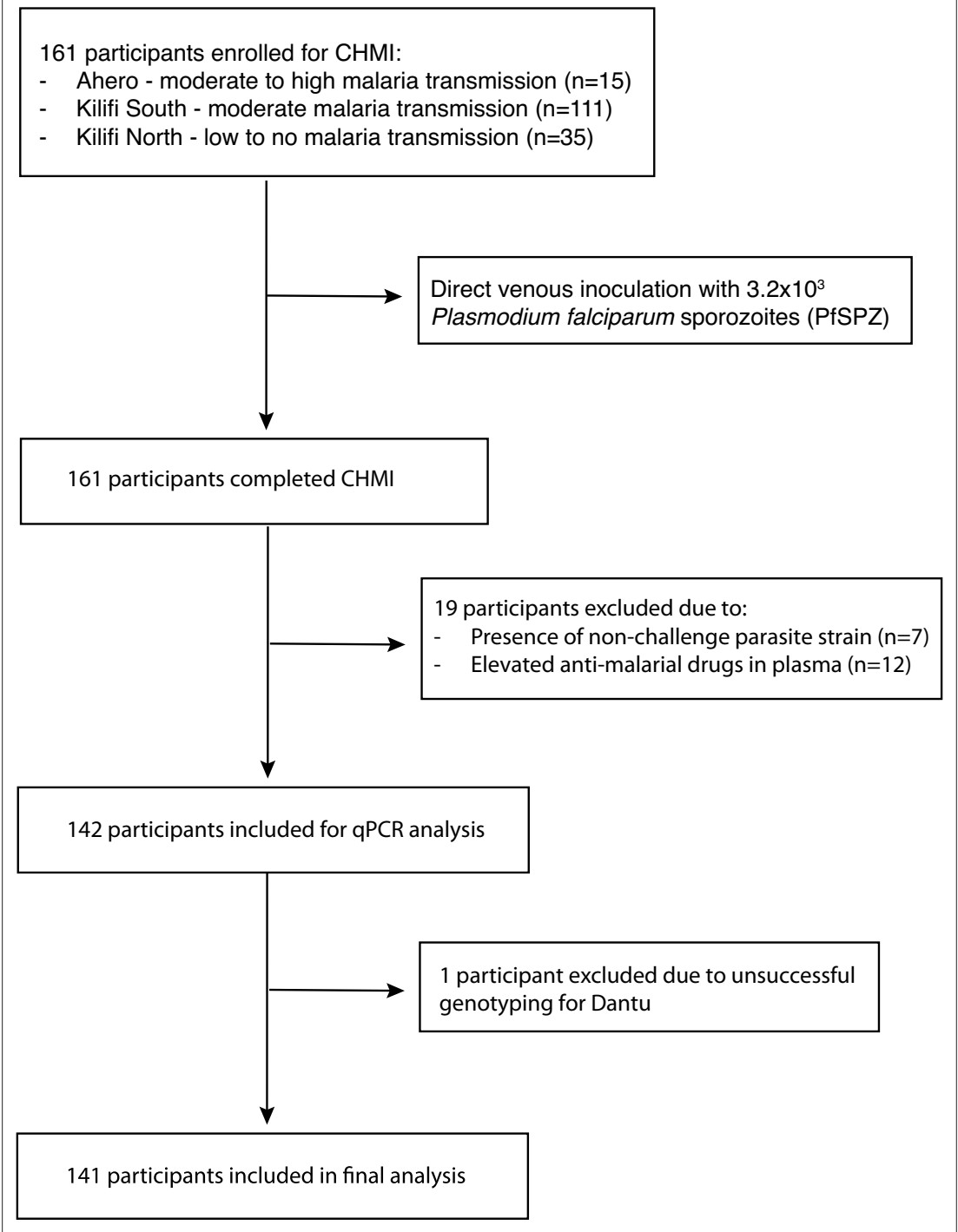

**Figure 1.** Study design and participant recruitment.

## Peak parasitaemias were lower in Dantu-carrying participants

The proportion of participants who became PCR-positive at any parasitaemia was similar at 86/111 (77.5%) in the non-Dantu and 20/27 (74.1%) and 2/3 (66.7%) among Dantu heterozygotes and homozygotes, respectively (p=0.745) (*Figure 4*). However, maximum parasitaemias were considerably higher in non-Dantu than in Dantu-carrying individuals. Peak parasitaemias reached 9694 parasites/µl in the non-Dantu group in comparison with 411 parasites/µl in the Dantu heterozygotes and only 3 parasites/µl among the Dantu homozygotes (non-Dantu vs. Dantu heterozygotes p=0.028; non-Dantu vs.

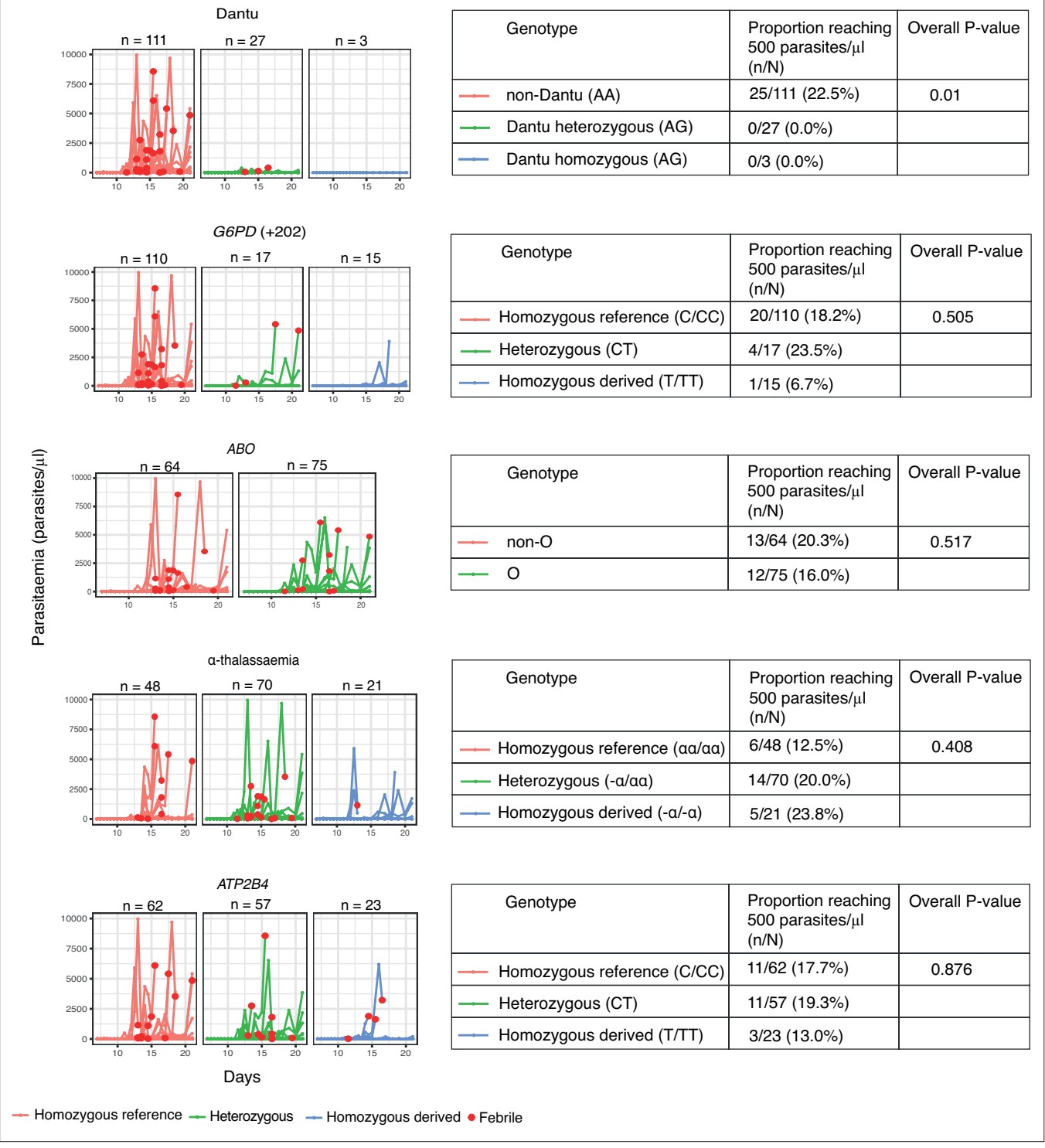

**Figure 2.** The impact of human genotype on parasite growth. Following inoculation of volunteers with Pf sporozoites, parasitaemia was monitored by quantitative PCR (y-axis) over the full duration of the study (x-axis). The different panels indicate the specific variants that were studied. The red dots indicate only the individuals that exhibited febrile symptoms and met treatment criteria. The different genotype groups for all the variants are categorised as 'Homozygous reference' for individuals with two copies of the reference allele (red lines), 'Heterozygous' for individuals with one copy of the reference allele and one copy of the derived allele (green lines), and 'Homozygous derived' for individuals with two copies of the derived allele (blue lines). αα/αα, no α-thalassaemia; −α/αα, heterozygous α-thalassaemia; −α/−α, homozygous α-thalassaemia. * For G6PD, male and female were combined as C/CC = normal (wild type hemizygous males and homozygous females), CT = carrier females, and T/TT = G6PD-deficient hemizygous males and homozygous females. All volunteers were typed for all variants, and any one individual may carry a mixture of genotypes – the potential

*Figure 2 continued on next page*

*Figure 2 continued*

confounding effect of this was controlled for in multivariate analysis. The tables adjacent to each plot show the results from Fisher's exact tests investigating differences in the proportion of participants that reached the pre-defined treatment threshold of 500 parasites/µl (n) compared to the total number within each genotype category (N).

The online version of this article includes the following source data for figure 2:

**Source data 1.** Related to *Figure 2*.

**Source data 2.** Related to *Figure 2* table.

**Table 1.** The proportion of participants reaching the pre-defined treatment parasitaemia threshold of 500 parasites/µl by genotype category.

| Variant | Genotype | n/N | % | p value overall | p value homozygous reference vs. heterozygous | p value homozygous reference vs. homozygous derived |
|---|---|---|---|---|---|---|
| Dantu rs186873296 | Non-Dantu (AA) | 25/111 | 22.5 | 0.01 | 0.004 | 1 |
| | Heterozygous (AG) | 0/27 | 0.0 | | | |
| | Dantu homozygous (GG) | 0/3 | 0.0 | | | |
| *G6PD* +202 rs1050828 | Homozygous reference (C/CC) | 20/110 | 18.2 | 0.505 | 0.739 | 0.463 |
| | Heterozygous (CT) | 4/17 | 23.5 | | | |
| | Homozygous derived (T/TT) | 1/15 | 6.7 | | | |
| *ABO* rs8176719 | Non-O | 13/64 | 20.3 | 0.517 | 0.517 | - |
| | O | 12/75 | 16.0 | | | |
| α-thalassaemia | Homozygous reference (αα/αα) | 6/48 | 12.5 | 0.408 | 0.328 | 0.29 |
| | Heterozygous (−α/αα) | 14/70 | 20.0 | | | |
| | Homozygous derived (−α/−α) | 5/21 | 23.8 | | | |
| *ATP2B4* rs4951074 | Homozygous reference (GG) | 11/62 | 17.7 | 0.876 | 1 | 0.749 |
| | Heterozygous (AG) | 11/57 | 19.3 | | | |
| | Homozygous derived (AA) | 3/23 | 13.0 | | | |

n = the number of participants that reached the pre-defined treatment threshold of 500 parasites/µl and were treated; N = the total number within each genotype category; αα/αα, no α-thalassaemia; −α/αα, heterozygous α-thalassaemia; −α/−α, homozygous α-thalassaemia. * For G6PD, male and female were combined as C/CC = normal (wild type hemizygous males and homozygous females), CT = carrier females, and T/TT = G6PD-deficient hemizygous males and homozygous females. We used the Fisher's exact test to investigate differences in the proportions of individuals that reached the pre-defined treatment threshold of 500 parasites/µl, both for global comparisons across genotype groups, and for separate comparisons between genotype pairs, where the proportion of individuals that reached the treatment threshold of 500 parasites/µl in the heterozygous- and homozygous -derived genotypes were compared to the homozygous reference genotype.

The online version of this article includes the following source data for table 1:

**Source data 1.** Related to *Table 1*.

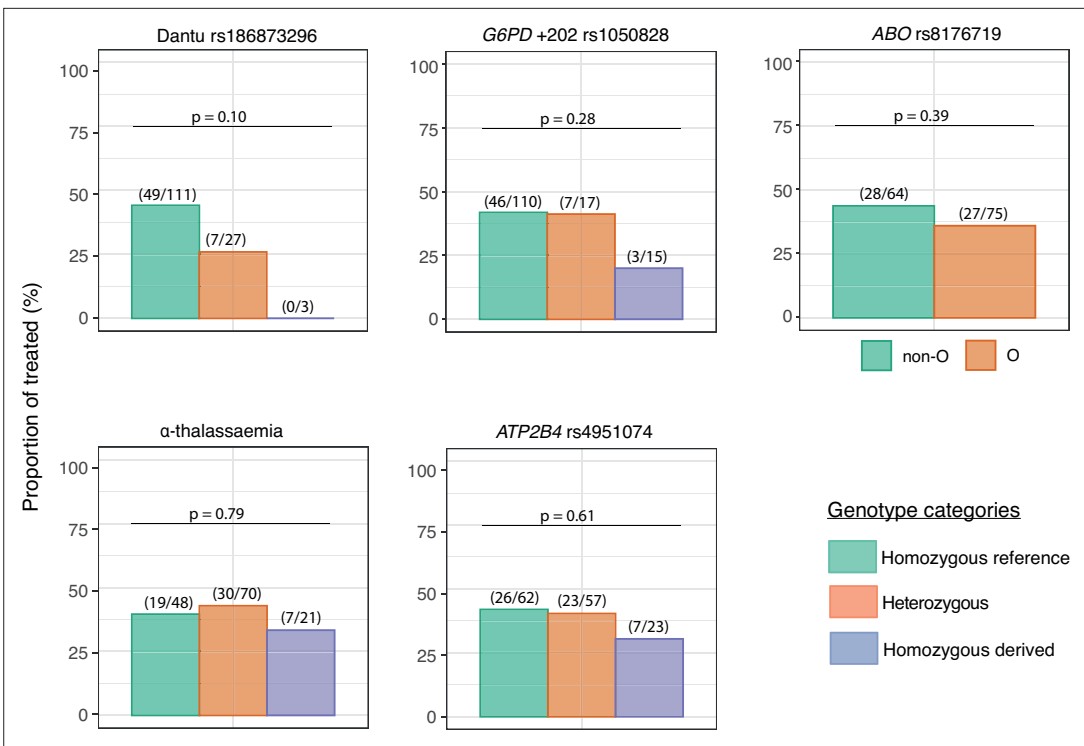

**Figure 3.** The impact of each gene variant on the requirement for malaria treatment.

The proportion of individuals in each genotype category that required treatment over the course of the controlled human malaria infection (CHMI) study is shown on the y-axis. The number of treated individuals out of the total number in each genotype group is given in parenthesis above the bar graphs, while the p values from the Fisher's exact tests comparing the differences in proportions of individuals that required treatment across genotype groups are also given above the bar graphs.

The online version of this article includes the following source data and figure supplement(s) for figure 3:

**Source data 1.** Related to *Figure 2*.

**Figure supplement 1.** No differences in anti-schizont antibody levels were found across Dantu genotype groups.

**Figure supplement 1—source data 1.** Related to *Figure 3—figure supplement 1*.

---

Dantu homozygotes p=0.141; and non-Dantu vs. Dantu heterozygotes and homozygotes combined p=0.009) (*Figure 4*). Similarly, the median parasitaemia among those who did become PCR-positive was 112 parasites/µl in the non-Dantu, 13 parasites/µl in the Dantu heterozygous, and 2 parasites/µl in the Dantu homozygous groups, respectively, although these differences did not reach statistical significance (non-Dantu vs. Dantu heterozygotes p=0.108; non-Dantu vs. Dantu homozygotes p=0.256; and non-Dantu vs. combined Dantu heterozygotes and homozygotes p=0.068) (*Figure 4*).

## Time to treatment was significantly longer in Dantu-carrying than non-Dantu individuals

Among the participants who did receive treatment, the time to treatment was significantly longer among Dantu-carrying than non-Dantu individuals. While the univariate comparisons across genotype groups, performed using the Log-Rank test in the Kaplan–Meier survival curves, did not reach statistical significance (*Figure 5a*), the multivariate Cox regression analysis with adjustments for other malaria-protective variants, anti-schizont antibody concentration, and location of residence showed that time to treatment was significantly longer among Dantu-carrying than non-Dantu individuals, the overall hazard ratio being 0.39 (CI 0.17–0.87; p=0.022) (*Figure 5b*). A dose-dependent effect of the Dantu genotype was also seen as none of the three Dantu homozygotes required treatment and the time to treatment was significantly longer in Dantu heterozygous than in non-Dantu individuals (HR = 0.41, p=0.042) (*Figure 5b*).

**Table 2.** The numbers and frequencies of individuals who received treatment before day 21 by genotypic category.

We compared the proportion of individuals that received treatment both across all genotype groups and in pairwise comparisons between pairs of genotype groups using the Fisher's exact test. In pairwise analyses, we compared, separately, the proportion of treated individuals in the heterozygous- and homozygous-derived genotypes to the homozygous reference genotype. n is the number of participants that were treated; N is the total number within the genotypic category; αα/αα, no α-thalassaemia; -α/αα, heterozygous α-thalassaemia; -α/-α, homozygous α-thalassaemia. * For G6PD, male and female were combined as C/CC = normal (wild type hemizygous males and homozygous females), CT = carrier females, and T/TT = G6PD-deficient hemizygous males and homozygous females.

| Variant | Genotype | n/N | % | Overall p value | Homozygous reference vs. heterozygous | Homozygous reference vs. homozygous derived |
|---|---|---|---|---|---|---|
| Dantu rs186873296 | Non-Dantu (AA) | 49/111 | 44.1 | 0.10 | 0.13 | 0.26 |
| | Heterozygous (AG) | 7/27 | 25.9 | | | |
| | Dantu homozygous (GG) | 0/3 | 0.0 | | | |
| G6PD +202 rs1050828* | Homozygous reference (C/CC) | 46/110 | 41.8 | 0.28 | 1.00 | 0.16 |
| | Heterozygous (CT) | 7/17 | 41.2 | | | |
| | Homozygous derived (T/TT) | 3/15 | 20.0 | | | |
| ABO rs8176719 | Non-O | 28/64 | 43.8 | 0.39 | - | 0.39 |
| | O | 27/75 | 36.0 | | | |
| α-thalassaemia | Homozygous reference (αα/αα) | 19/48 | 39.6 | 0.79 | 0.85 | 0.79 |
| | Heterozygous (−α/αα) | 30/70 | 42.9 | | | |
| | Homozygous derived (−α/−α) | 7/21 | 33.3 | | | |
| ATP2B4 rs4951074 | Homozygous reference (GG) | 26/62 | 41.9 | 0.61 | 1.00 | 0.45 |
| | Heterozygous (AG) | 23/57 | 40.4 | | | |
| | Homozygous derived (AA) | 7/23 | 30.4 | | | |

The online version of this article includes the following source data for table 2:

**Source data 1.** Related to *Table 2*.

## No significant impacts were seen with regard to any of the outcomes under study for any of the remaining RBC polymorphisms

While our primary analysis focused on the Dantu genotype, individuals in malaria-endemic regions often carry more than one malaria protection-associated genotype (*Ndila et al., 2018*). We therefore typed individuals for α-thalassaemia, blood group O, G6PD deficiency and *ATP2B4* alleles, and carried out the same set of analysis as above for each genotype. Each of these genotypic groups also included, by random chance, individuals of different genotypes for the other variants of interest. As the genotyping was conducted at the end of the study, and the study included only a relatively small number of individuals overall, it was not possible to limit each group to those who were reference homozygotes for all the other variants of interest. Instead, we used multivariate analyses to check that any differences seen on univariate analysis were not explained by the presence of other variants or confounders. No significant differences were seen in any of the study outcomes between the different

**Table 3.** The association between each gene variant and the requirement for treatment after challenge.

The association between each variant genotype and the treated or untreated categorical outcome was analysed by multivariate logistic regression using additive models for each variant, where each variant genotype was coded as zero, one, or two copies of the homozygous-derived allele. Pairwise analysis compared the treatment outcomes for the heterozygous- and homozygous-derived genotypes to the homozygous reference genotype. Adjustments were made for other gene variants, anti-schizont antibody levels, and location of residence.

| Variant | Overall comparison across genotype groups | | | Homozygous reference vs. heterozygous | | | Homozygous reference vs. homozygous derived | | |
|---|---|---|---|---|---|---|---|---|---|
| | Odds ratios | 95% CI | p-value | Odds ratios | 95% CI | p value | Odds ratios | 95% CI | p value |
| Dantu rs186873296 | 0.17 | 0.04–0.55 | 0.007 | 0.20 | 0.04–0.83 | 0.039 | 0 | NA – Inf | 0.990 |
| G6PD +202 rs1050828 | 0.60 | 0.28–1.19 | 0.157 | 1.71 | 0.41–6.59 | 0.442 | 0.17 | 0.02–0.97 | 0.074 |
| ABO rs8176719 | 0.40 | 0.15–1.03 | 0.064 | 0.54 | 0.18–1.56 | 0.259 | - | - | - |
| α-thalassaemia | 0.91 | 0.46–1.73 | 0.766 | 0.97 | 0.31–3.04 | 0.957 | 0.87 | 0.18–3.22 | 0.758 |
| ATP2B4 rs4951074 | 0.84 | 0.43–1.63 | 0.619 | 0.53 | 0.17–1.58 | 0.266 | 0.57 | 0.10–2.58 | 0.491 |

The online version of this article includes the following source data for table 3:

**Source data 1.** Related to *Table 3*.

genotype groups (*Figures 2, 3 and 5*, *Tables 1–3*). As noted above, the impact of Dantu on malaria treatment was independent of the presence or absence of these other genetic variants in multivariate analysis.

## Discussion

Through the analysis of data from a CHMI study with PfSPZ Challenge injection conducted in Kenyan adults, we have shown that the Dantu blood group is associated with prevention of parasite growth *in vivo*. While more than 20% of Dantu-negative volunteers developed bloodstream malaria infections that reached a pre-defined threshold of 500 parasites/µl following controlled PfSPZ administration, this threshold was not reached by any of the 30 Dantu-positive volunteers. This is the first time that Dantu has been shown to protect against early-stage malaria infections. Our study suggests that Dantu protects against severe and complicated malaria (*Ndila et al., 2018*; *Band et al., 2015*) by preventing the disease from becoming established in its earliest phase. This is consistent with recent *in vitro* observations that have demonstrated a link between Dantu genotype and susceptibility to red blood cell invasion by *P. falciparum* merozoites, which we previously predicted would lead to reduced parasite growth *in vivo* (*Kariuki et al., 2020*). That mechanistic study revealed that the Dantu genotype protects red blood cells from invasion by increasing their surface tension, which reduces the ability of merozoites to deform their surface and hence productively invade. Critically, this tension difference was greater in Dantu homozygotes than heterozygotes, as was the reduction in invasion efficiency, supporting a dose-dependent protective effect. This dose-dependency was similarly reflected in our current *in vivo* study, as while 44.1% of the Dantu-negative volunteers developed symptoms that precipitated malaria treatment, this occurred in only 25.9% of Dantu heterozygous and in none of the three (0%) Dantu homozygous volunteers. Similarly, the maximum parasitaemia observed was considerably higher in the non-Dantu than in the Dantu heterozygous and homozygous participants.

In contrast to CHMI studies in malaria-naïve populations, where most participants typically develop clinical malaria (*Church et al., 1997*), less than one-fifth of the participants in this study reached a threshold of >500 parasites/µl. This is probably because most of the participants in our study were residents of malaria-endemic communities and will therefore have been partially immune to the disease. Indeed, in previous analyses of data from the same study we have shown that infection outcome

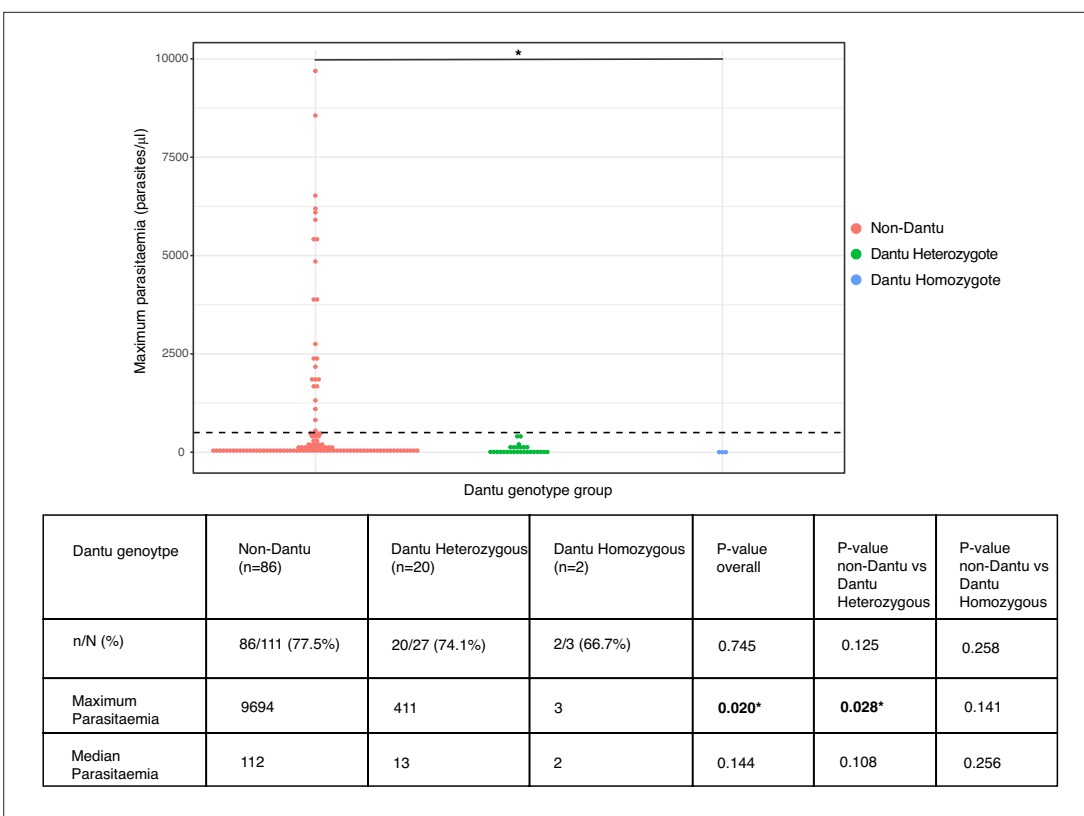

| Dantu genoytpe | Non-Dantu (n=86) | Dantu Heterozygous (n=20) | Dantu Homozygous (n=2) | P-value overall | P-value non-Dantu vs Dantu Heterozygous | P-value non-Dantu vs Dantu Homozygous |
|---|---|---|---|---|---|---|
| n/N (%) | 86/111 (77.5%) | 20/27 (74.1%) | 2/3 (66.7%) | 0.745 | 0.125 | 0.258 |
| Maximum Parasitaemia | 9694 | 411 | 3 | **0.020*** | **0.028*** | 0.141 |
| Median Parasitaemia | 112 | 13 | 2 | 0.144 | 0.108 | 0.256 |

**Figure 4.** Peak parasitaemias were lower in Dantu variant carriers.

Maximum parasitaemia values for individuals across Dantu genotype groups, with dashed line indicating the treatment threshold of 500 parasites/µl. The table below the figure shows the numbers and frequencies of individuals in each genotype category that were PCR-positive over the course of the controlled human malaria infection (CHMI) study. n = the number of participants that were PCR-positive; N = the total number within the genotype category. Statistical comparisons of proportions of PCR-positive individuals across genotype groups and pairwise comparisons between genotype groups were performed using the Fisher's exact test. Statistical comparisons of maximum and median parasitaemia between genotype groups were performed using the Kruskal–Wallis test, and post-hoc Dunn's test for pairwise differences between the genotype groups.

The online version of this article includes the following source data for figure 4:

**Source data 1.** Related to *Figure 4*.

**Source data 2.** Related to *Figure 4* table.

was attributable to the degree of prior exposure as estimated by the titre of anti-schizont antibodies (*Kapulu et al., 2022*), as well as other measures of exposure including the antibody-dependent phagocytosis of both ring-infected and uninfected erythrocytes from parasite cultures (*Musasia et al., 2022*) and the breadth of antibodies to *P. falciparum* Variant Surface Antigens (*Kimingi et al., 2022*). As such, Dantu genotype was therefore clearly not the only factor at play in determining the clinical outcome in this study. However, our multivariate analysis adjusted for these factors (*Kapulu et al., 2022*) as well as other malaria-protective genetic variants, and the significant association in that analysis underscores the strong protective effect conferred by Dantu. There were also no differences observed in anti-schizont antibody levels across Dantu genotype groups, suggesting that differences in pre-existing anti-malaria immunity between Dantu and non-Dantu cannot explain the differences seen in this study.

The protective impact of Dantu against *in vivo* parasite growth was in stark contrast to that of the other genetic factors under study. Although consistent evidence has been found for protective effects against severe malaria by G6PD deficiency, blood group O, the rs4951074 allele in *ATP2B4* and α+-thalassaemia in numerous previous studies (*Malaria Genomic Epidemiology Network, 2014*; *Taylor et al., 2012*), none of these polymorphisms had any significant impacts on any of the outcomes

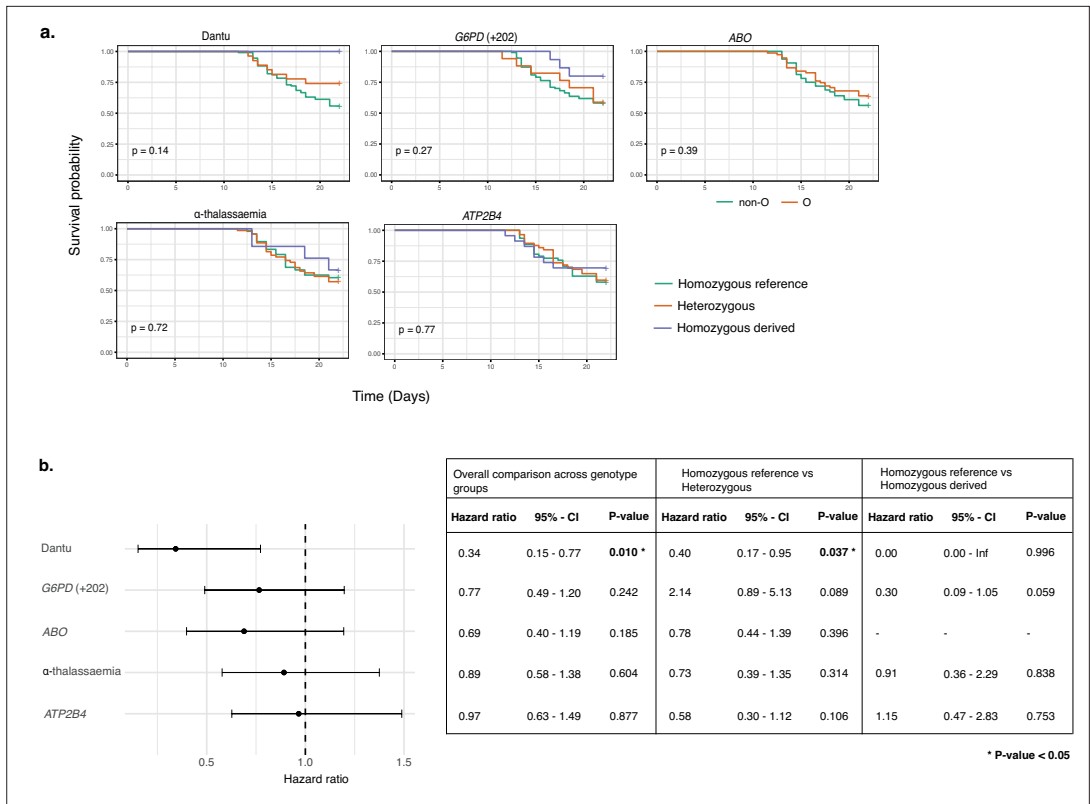

**Figure 5.** Time to treatment was longer in Dantu variant carriers.

The impact of each gene variant on time to treatment was analysed by (**a**) Kaplan–Meier survival curves, with univariate comparisons across genotype groups performed using the Log-Rank test and (**b**) multivariate Cox regression models, with each variant genotype coded as zero, one, or two copies of the homozygous derived allele in an additive model, adjusting for the other four malaria-protective variants, anti-schizont antibody concentration, and location of residence. Pairwise analysis compared the time to treatment in the heterozygous- and homozygous-derived genotypes to the homozygous reference genotype.

The online version of this article includes the following source data for figure 5:

**Source data 1.** Related to *Figure 5a*.

**Source data 2.** Related to *Figure 5b*.

**Source data 3.** Related to *Figure 5b* table.

under investigation in this study. This is probably because, unlike Dantu, none of these conditions have a clearly established impact on merozoite red cell invasion, but instead influence the progress of malaria once the disease has become established. For example, recent studies have shown that $\alpha^+$-thalassaemia has no effect on either red blood cell invasion (**Williams et al., 2002**) or the development of uncomplicated malaria (**Taylor et al., 2012**; **Wambua et al., 2006**), but protects instead against the development of severe and complicated disease through mechanisms that include the reduced expression of red cell surface antigens that result in cytoadhesion (**Krause et al., 2012**; **Opi et al., 2014**). Similarly, it had no apparent impact on infectivity in a previous, smaller, CHMI study using PfSPZ Challenge, conducted in Tanzania (**Shekalaghe et al., 2014**). Because clinical guidelines mean that controlled human malaria challenge studies only ever reach relatively low-density parasitaemias, they may not adequately capture the impacts of genetic factors that influence the later, more severe stages of malaria disease. However, this study shows that they can be helpful in pointing towards the pathways leading to such outcomes for further study by other methods.

In conclusion, this study reveals the power of CHMI studies to deconvolute the malaria-protective effects of naturally occurring human genetic variants, establishes for the first time that the Dantu blood group provides strong protection against *in vivo* parasite growth, and emphasises the potential of Dantu-phenocopying interventions to limit *P. falciparum* growth *in vivo*.

## Additional information

### Competing interests

Stephen L Hoffman: SLH and members of the CHMI-SIKA Study Team (YA, PFB, ERJ, TLR, and BKLS; see Author Information) are salaried, full-time employees of Sanaria, the manufacturer of Sanaria PfSPZ Challenge. The authors have no additional financial interests. Melissa C Kapulu: Melissa C Kapulu has received grants from UKRI MRC UK and Wellcome Trust and has received travel expenses for the HIC-VAC Annual Meeting, Greenwood Africa Prize and WHO working group meeting on challenge studies. The author has no other competing interests to declare. The other authors declare that no competing interests exist.

### Funding

| Funder | Grant reference number | Author |
|---|---|---|
| Wellcome Trust | 107499 | Melissa C Kapulu<br>Philip Bejon |
| Wellcome Trust | 216444/Z/19/Z | Silvia N Kariuki |
| Wellcome Trust | 202800/Z/16/Z | Thomas N Williams |
| Wellcome Trust | 220266/Z/20/Z | Julian C Rayner |

The funders had no role in study design, data collection and interpretation, or the decision to submit the work for publication. For the purpose of Open Access, the authors have applied a CC BY public copyright license to any Author Accepted Manuscript version arising from this submission.

### Author contributions

Silvia N Kariuki, Conceptualization, Data curation, Formal analysis, Investigation, Visualization, Methodology, Writing - original draft, Writing – review and editing; Alexander W Macharia, Johnstone Makale, Investigation, Methodology; Wilfred Nyamu, Investigation; Stephen L Hoffman, Conceptualization, Resources, Methodology, Writing – review and editing; Melissa C Kapulu, Conceptualization, Resources, Investigation, Methodology, Project administration, Writing – review and editing; Philip Bejon, Conceptualization, Resources, Formal analysis, Supervision, Methodology, Writing – review and editing; Julian C Rayner, Conceptualization, Supervision, Writing – review and editing; Thomas N Williams, Conceptualization, Resources, Supervision, Methodology, Writing – review and editing

### Author ORCIDs

Silvia N Kariuki http://orcid.org/0000-0003-0801-5285
Julian C Rayner http://orcid.org/0000-0002-9835-1014
Thomas N Williams http://orcid.org/0000-0003-4456-2382

### Ethics

Clinical trial registration: The study is registered on ClinicalTrials.gov (NCT02739763).
The study was approved by both the KEMRI Scientific and Ethics Review Unit (protocol KEMRI/SERU/CGMR-C/029/3190) in Kenya, and the University of Oxford Tropical Research Ethics Committee (OxTREC; protocol 2-16) in the UK. The study was conducted based on good clinical practice (GCP) following the principles of the Declaration of Helsinki.

### Decision letter and Author response

Decision letter https://doi.org/10.7554/eLife.83874.sa1
Author response https://doi.org/10.7554/eLife.83874.sa2

## Additional files

### Supplementary files

- MDAR checklist
- Reporting standard 1. STREGA checklist.

## Data availability

All data generated or analysed during this study are included in the manuscript and supporting file; Source Data files have been provided for Figures 2, 3, 4, and 5 and Tables 1, 2 and 3.

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

# Appendix 1

## Members of the CHMI-SIKA Study Team

**Centre for Geographic Medicine Research (Coast), Kenya Medical Research Institute-Wellcome Trust Research Programme, Kilifi, Kenya:** Abdirahman I Abdi, Philip Bejon, Zaydah de Laurent, Mainga Hamaluba, Domtila Kimani, Rinter Kimathi, Kevin Marsh, Sam Kinyanjui, Khadija Said Mohammed, Moses Mosobo, Janet Musembi, Jennifer Musyoki, Michelle Muthui, Jedidah Mwacharo, Kennedy Mwai, Joyce M Ngoi, Omar Ngoto, Patricia Njuguna, Irene Nkumama, Francis Ndungu, Dennis Odera, Donwilliams Omuoyo, Faith Osier, Edward Otieno, Jimmy Shangala, James Tuju, Juliana Wambua, Thomas N Williams

**Sanaria Inc, Rockville, MD, United States:** Yonas Abebe, Peter F Billingsley, Stephen L Hoffman, Eric R James, Thomas L Richie, B Kim Lee Sim

**Centre for Tropical Medicine and Global Health, Nuffield Department of Medicine, University Oxford, Oxford, United Kingdom:** Sam Kinyanjui, Kevin Marsh

**Department of Pathology, University of Cambridge, Cambridge, United Kingdom:** Peter C Bull

**Pwani University, P. O. Box 195-80108, Kilifi, Kenya:** Sam Kinyanjui, Cheryl Kivisi

**Epidemiology and Biostatistics Division, School of Public Health, University of the Witwatersrand, Johannesburg, South Africa:** Kennedy Mwai

**Centre for Infectious Diseases, Heidelberg University Hospital, Heidelberg, Germany:** Irene Nkumama, Dennis Odera, Faith Osier

**Center for Research in Therapeutic Sciences, Strathmore University, Nairobi, Kenya:** Bernhards Ogutu, Fredrick Olewe, John Ong'echa

**Center for Research in Therapeutic Sciences, Strathmore University, Nairobi, Kenya:** Bernhards Ogutu, Fredrick Olewe, John Ong'echa

