## [Editor Report]

The large genetic association studies conducted in East Africa have shown that the Dantu blood group provides substantial protection against severe malaria since it increases the surface tension of red blood cells making it harder for malaria parasites to invade. In this important work, the authors show that parasite growth is indeed restricted *in vivo* by testing this hypothesis in adult Kenyan volunteers infected with *P. falciparium* under careful monitoring. They were able to show convincingly that indeed, parasite growth was reduced amongst Dantu adults.

---

## [Decision Letter]

**Decision letter after peer review:**

Thank you for submitting your article "The Dantu blood group provides high level protection against uncomplicated malaria: evidence from a Controlled Human Malaria Infection study" for consideration by *eLife*. Your article has been reviewed by 2 peer reviewers, and the evaluation has been overseen by a Reviewing Editor and Jos van der Meer as the Senior Editor. The following individuals involved in review of your submission have agreed to reveal their identity: Wiebke Nahrendorf (Reviewer #1); Nicholas J White (Reviewer #2). The reviewers have some suggestions to take into account when you resubmit.

*Reviewer #1 (Recommendations for the authors):*

Nice short paper showing that Dantu prevents parasite growth *in vivo*. I think it is important to be really clear that the mechanism of protection is that red cells are rigid so parasites cannot get in. This makes a lovely clear-cut story so I would recommend changing the title to something like "The Dantu blood group prevents parasite growth *in vivo*." – simple (and exciting!).

At present the title and some of the wording around uncomplicated malaria is a little misleading: of course if there are no malaria parasites no malaria symptoms are experienced. "Protection from uncomplicated malaria" would to me imply that at identical parasite densities Dantu individuals experience less symptoms than non-Dantu – but this is of course not what the paper shows (and the study would not be powered to do so).

Below a few comments on how the data is presented – my recommendation is to incorporate the key bits of information from Tables in the Figures as it is hard to interpret dissociated information. Also more informative labels and legends (include details of statistical test and n) in the Figures would help them be more self-explanatory methods: more details for genotyping – primer sequences etc. what is the reference allele? more details & references for R packages used for multivariate analysis. define fixed and random effects and all response variables.

Figure 2: More intuitive labelling: in text referred to as "non Dantu" "Dantu heterozygotes" and "Dantu homozygotes" – keep this in Figures for consistency (also for the other red cell polymorphisms). Why does the red group look different between the different polymorphisms? Does that mean e.g. G6PD contains Dantu heterozygotes/homozygotes? Should all polymorphisms be excluded? It was not clear to me what the control group is ("homozygous reference" is not mentioned in methods or elsewhere in text).

To help interpretation the number of participants in each graph should be included. My suggestion would be to incorporate the information of Table 1 into Figure 2: n could be indicated above each existing parasite growth curve graph, a small Figure (a bit like Figure 3) showing proportion of individuals that reached the parasitaemia threshold would fit next to it and could include p values. Bit more information in the legend would help the Figure to be more self-explanatory: "Following inoculation of volunteers with Pf sporozoites parasitaemia was monitored by qPCR. Here split by red cell polymorphisms. The control (red) is xyz…".

I was also not quite clear on the red dots: do they indicate all treated participants – so when they either crossed 500 p/ul threshold or had any parasites + symptoms? In which case why are there some very high parasitaemias with no red dot? Or just febrile participants? Consider if this is the right place for this information given that this Figure is about parasite growth.

Table 1: My suggestion would be to incorporate the key info into corresponding Figure 2. The heading needs to make it to clear that this is related to parasite growth – the "predefined threshold" is 500 parasites/ul. I also think n is the number of participants being treated (not "left untreated" as stated). Please check the p values and indicate the statistical test/multiple testing correction used in legend – how can Dantu p-value Homozygous Reference vs Homozygous Derived be 1…? I imagine a Wilcoxon rank sum exact test (two-tailed) would be most appropriate.

Figure 3: Group labelling like Figure 2 (e.g. "non Dantu"). Heading a bit misleading – the outcome of treatment is not considered. If n in Figure 2 does not need to be repeated here above each bar. Instead incorporate p values from Table 2. In text explain what variables were adjusted for and provide a formula for your multivariate regression analysis: which dependent and independent variables did you use?

Table 2: incorporate information in Figure 3. Like in Table 1 n is the number of treated (not untreated) participants.

Table 3: Again I think "treatment outcome" in title is misleading. The multivariate model and which variables were adjusted for needs to be clearly stated in legend and Results section (currently in methods only but with not enough details).

Figure 4: descriptive title matching subheading in results might be better: "Lower peak parasitaemia in Dantu-carrying participants". Axis labels and dots too small. Information from Table 4 could again be easily contained within this Figure.

Table 4: transfer key information to Figure 4. (or include a small Table in Figure 4)

Figure 5: Group labelling like Figure 2/3, axis labels too small. Condense the information provided within the table or at least highlight the key bits of information.

Figure 5 S1: Group labelling like Figure 2/3/5. I think this should go into Figure 5.

*Reviewer #2 (Recommendations for the authors):*

This is a very nice study. The result is clear and the data on the other genetic human polymorphisms are also very interesting!

Here are a few points to consider:

1. qPCR is a lot less accurate than microscopy. Was the primary endpoint determined by microscopy also and if so how did the two methods compare?

2. Obvious questions, but I presume that other protective genetic polymorphisms were not linked, and the Dantu blood groups were not clustered in an ethnic group with more "immunity" i.e. from a focus of higher transmission? I think both were addressed -but wish to check.

3. Ideally growth rates would have been calculated but it looks as if the data are too noisy. Is that correct?

4. Have you counted merozoites per schizont and asexual cycle lengths in in-vitro cultures of *P. falciparum* infected Dantu blood group red cells to confirm there is no growth inhibition ?

5. How were blood samples screened for antimalarial drugs?

---

## [Author Response]

Reviewer #1 (Recommendations for the authors):Nice short paper showing that Dantu prevents parasite growth in vivo. I think it is important to be really clear that the mechanism of protection is that red cells are rigid so parasites cannot get in. This makes a lovely clear-cut story so I would recommend changing the title to something like "The Dantu blood group prevents parasite growth in vivo." – simple (and exciting!).

We thank the reviewer for this helpful comment. Following the reviewer’s suggestion, we have now changed the title to: The Dantu blood group prevents parasite growth *in vivo*: evidence from a Controlled Human Malaria Infection study.

At present the title and some of the wording around uncomplicated malaria is a little misleading: of course if there are no malaria parasites no malaria symptoms are experienced. "Protection from uncomplicated malaria" would to me imply that at identical parasite densities Dantu individuals experience less symptoms than non-Dantu – but this is of course not what the paper shows (and the study would not be powered to do so).

We agree that our study does not compare symptomatology across genotype groups. We have now removed wording about uncomplicated malaria from the title and the rest of the manuscript as suggested by this reviewer.

Below a few comments on how the data is presented – my recommendation is to incorporate the key bits of information from Tables in the Figures as it is hard to interpret dissociated information. Also more informative labels and legends (include details of statistical test and n) in the Figures would help them be more self-explanatory methods: more details for genotyping – primer sequences etc. what is the reference allele? more details & references for R packages used for multivariate analysis. define fixed and random effects and all response variables.

Thank you for this helpful comment. We have now added details of the primer/probe sequences used to genotype the Dantu marker SNP, as well as the reference allele, to the Methods section of the revised paper on page 6. The multivariate analysis was performed with the glm function in the stats package (version 3.6.2), with the response variables being the Dantu marker SNP, other malaria-protective SNPs, anti-schizont antibody levels and area of residence. We have now included details of this and other packages within R that we used in our analyses to the statistical analysis section on page 7.

Figure 2: More intuitive labelling: in text referred to as "non Dantu" "Dantu heterozygotes" and "Dantu homozygotes" – keep this in Figures for consistency (also for the other red cell polymorphisms).

Thank you for this. We previously used “homozygous reference”, “heterozygous” and “homozygous derived” as a generalised labelling for the genotype categories in each of the different red cell polymorphisms. In the light of this comment, we have now clarified these genotype categories in the legend to Figure 2, while also amending the description for the Dantu group within this Figure for ease of reference. We hope this addresses this point to the reviewer’s satisfaction.

Why does the red group look different between the different polymorphisms?

In this Figure, the group coloured in red are the homozygous reference groups for each gene under investigation. In all cases, this group contains the largest number of individuals, but the precise number of individuals will differ depending on the minor allele frequency of each specific gene – this explains why the red group looks different in each case.

Does that mean e.g. G6PD contains Dantu heterozygotes/homozygotes? Should all polymorphisms be excluded?

As noted above, in this Figure the groups coloured in red are the homozygous reference individuals for each gene under investigation. Each of these genotypic groups will also include (by random chance) individuals of different genotypes for the other variants of interest. As the genotyping was conducted at the end of the study, and the study included only a relatively small number of individuals overall, it was not possible to limit each group to those who were reference homozygotes for all the other variants of interest. Instead, we used multivariable analyses to check that any differences seen on univariate analysis were not explained by the presence of other variants or confounders; these multivariate analyses clearly showed that the effect of Dantu was not due to such confounders. We hope this addresses the reviewer’s query. We have amended the text to address this point on page 11 of the revised manuscript.

It was not clear to me what the control group is ("homozygous reference" is not mentioned in methods or elsewhere in text).

No “control” group was included in this specific analysis. The overall p-values presented in the table were derived using Fisher’s exact tests, that were used to test for differences in the proportions of individuals that reached the pre-defined treatment threshold of 500 parasites/ml across ALL genotype groups – i.e. a global rather than a paired approach. We have added an explanation about the statistical methods involved both in the footnote to Figure 2 and in the Methods section on page 7.

To help interpretation the number of participants in each graph should be included. My suggestion would be to incorporate the information of Table 1 into Figure 2: n could be indicated above each existing parasite growth curve graph, a small Figure (a bit like Figure 3) showing proportion of individuals that reached the parasitaemia threshold would fit next to it and could include p values. Bit more information in the legend would help the Figure to be more self-explanatory: "Following inoculation of volunteers with Pf sporozoites parasitaemia was monitored by qPCR. Here split by red cell polymorphisms. The control (red) is xyz…".

Thank you for this suggestion. We have now modified Figure 2 as suggested to include more information both on each graph and in the legend.

I was also not quite clear on the red dots: do they indicate all treated participants – so when they either crossed 500 p/ul threshold or had any parasites + symptoms? In which case why are there some very high parasitaemias with no red dot? Or just febrile participants? Consider if this is the right place for this information given that this Figure is about parasite growth.

The red dots indicate just the febrile participants (not all of them exhibited febrile symptoms even at the very high parasitaemias). We have now added an explanation about this point to the Figure legend.

Table 1: My suggestion would be to incorporate the key info into corresponding Figure 2. The heading needs to make it to clear that this is related to parasite growth – the "predefined threshold" is 500 parasites/ul. I also think n is the number of participants being treated (not "left untreated" as stated). Please check the p values and indicate the statistical test/multiple testing correction used in legend – how can Dantu p-value Homozygous Reference vs Homozygous Derived be 1…? I imagine a Wilcoxon rank sum exact test (two-tailed) would be most appropriate.

Due to the low sample sizes in the different genotype categories, we used the Fisher’s exact test to compare the proportions of individuals that reached the pre-defined treatment threshold of 500 parasites/ml (see the numbers in the table below). Both global comparisons across genotype groups and pairwise comparisons between genotype groups were done using the Fisher’s exact test. We did not correct for multiple testing in the pairwise analyses because we did separate comparisons of the proportion of individuals that reached the treatment threshold of 500 parasites/ml in the heterozygous and homozygous derived genotypes to the homozygous reference genotype. We have updated the footnote to Author response table 1 to explain the statistical methods that were used in better detail.

**Author response table 1. sa2table1:** 

Dantu genotype group	Below 500 parasites/ul	Above 500 parasites/ul	Sum
Non-Dantu (AA)	86	25	111
Dantu Heterozygote (AG)	27	0	27
Dantu Homozygote	3	0	3

Contingency table for maximum PCR values categorised as below 500 parasites/ul and above 500 parasites/ul

Figure 3: Group labelling like Figure 2 (e.g. "non Dantu"). Heading a bit misleading – the outcome of treatment is not considered. If n in Figure 2 does not need to be repeated here above each bar. Instead incorporate p values from Table 2. In text explain what variables were adjusted for and provide a formula for your multivariate regression analysis: which dependent and independent variables did you use?

Thank you for this. We have now added more information in this Figure about the numbers of individuals in each genotype group that required treatment, as well as the p-values from the Fisher’s exact tests comparing the differences in proportions of individuals that required treatment across genotype groups. The results from the multivariate regression are listed in Table 3. We have added further details to the Methods section on pages 7 and 8 about the Fisher’s exact test and multivariate regression models used.

Table 2: incorporate information in Figure 3. Like in Table 1 n is the number of treated (not untreated) participants.

Our apologies for this. We have corrected the information on the number of treated participants in the legend and included the p-values comparing the proportions of individuals requiring treatment across genotypes in Figure 3.

Table 3: Again I think "treatment outcome" in title is misleading. The multivariate model and which variables were adjusted for needs to be clearly stated in legend and Results section (currently in methods only but with not enough details).

We agree with this point. We have now edited the title and legend to make it clearer that it is the impact of each gene variant on treatment requirement that we are analysing. We have also added further details to the legend and the Results section on page 9 regarding the multivariate models used.

Figure 4: descriptive title matching subheading in results might be better: "Lower peak parasitaemia in Dantu-carrying participants". Axis labels and dots too small. Information from Table 4 could again be easily contained within this Figure.

We thank the reviewer for this helpful suggestion. We have now edited the subheading of Figure 4 and incorporated the information from Table 4 into this figure accordingly.

Table 4: transfer key information to Figure 4. (or include a small Table in Figure 4)

Thank you. We have now included a small Table in Figure 4.

Figure 5: Group labelling like Figure 2/3, axis labels too small. Condense the information provided within the table or at least highlight the key bits of information.Figure 5 S1: Group labelling like Figure 2/3/5. I think this should go into Figure 5.

We have now edited Figure 5 into a multi-part figure including Figure 5 S1.

Reviewer #2 (Recommendations for the authors):This is a very nice study. The result is clear and the data on the other genetic human polymorphisms are also very interesting!Here are a few points to consider:1. qPCR is a lot less accurate than microscopy. Was the primary endpoint determined by microscopy also and if so how did the two methods compare?

Because of the clinical guidelines around CHMI studies, participants need to be treated very early, meaning parasites only ever reach a relatively low density. In a previous publication (Bejon et al. Malaria J 2006, new Ref. # 16), parasite density measurements by qPCR and microscopy were compared using serially diluted parasite cultures of known parasitaemia. This study clearly showed that qPCR gives more accurate readings of parasite numbers at low densities, while thick blood films are less reproducible. As a consequence, we used qPCR as the primary endpoint measurement of parasitaemia in our current study, and it is also routinely used as the primary endpoint in CHMI studies conducted by other groups around the world (Refs. # 17 – 23). Thick blood smears were also performed but this was as an additional clinical precaution, with participants being treated if they became blood film positive at any density. We have added more information in the Methods section on page 5 to clarify this point, and added the references discussed here.

2. Obvious questions, but I presume that other protective genetic polymorphisms were not linked, and the Dantu blood groups were not clustered in an ethnic group with more "immunity" i.e. from a focus of higher transmission? I think both were addressed -but wish to check.

While all 3 Dantu homozygotes came from the same location (Kilifi South), as noted in the response to Reviewer 1, we compared anti-schizont antibody levels of all individuals to assess “immunity” and used the readings as a factor in our multivariate analysis. We found no differences in anti-schizont antibody levels, a correlate of pre-existing immunity to malaria, between the genetic groups. We have now included these data in Figure 3 —figure supplement 1.

3. Ideally growth rates would have been calculated but it looks as if the data are too noisy. Is that correct?

This is correct. Growth rates were calculated in a previous study (Kapulu et al. BMC Infect Dis, 2022; Ref. # 21), and they were indeed found to be too noisy to allow for meaningful further analysis. This is because, to a variable extent, all of the participants had been previously exposed to malaria infections and their growth rate measurement data varied very significantly as a result, compared to what you would expect among malaria-naïve individuals such as those used in CHMI malaria vaccine trials. Of course, it would be next to impossible to perform CHMI to explore malaria-protective genotypes, which are enriched in African populations, with completely malaria-naïve individuals. We have not changed the manuscript in this regard but would be happy to add a comment in the discussion if helpful.

4. Have you counted merozoites per schizont and asexual cycle lengths in in-vitro cultures of *P. falciparum* infected Dantu blood group red cells to confirm there is no growth inhibition ?

This is a good suggestion but, unfortunately, because we used qPCR, rather than microscopy, as our primary method for tracking parasitaemia, we are not able to count merozoites per schizont and asexual cycle lengths in infected Dantu variant red cells in this study. This would require further *in vitro* experiments rather than this study’s *in vivo* experimental design, where for clinical reasons, treatment intervention guidelines mean that parasitaemia levels are never sufficiently high to generate such data.

5. How were blood samples screened for antimalarial drugs?

Antimalarial drugs were measured retrospectively in all volunteers at two time points, the day before the challenge and eight days after challenge. Their plasma samples were tested in two independent laboratories: Strathmore University in Nairobi, Kenya, which measured sulfadoxine, pyrimethamine, and chloroquine levels; and Mahidol Oxford Tropical Medicine Research Unit in Bangkok, Thailand, which measured artemether and dihydroartemisinin concentrations. All drug concentrations were measured by liquid chromatography-tandem mass spectrometry. These methods were detailed in a previous publication (Kapulu et al. JCI Insight 2021; Ref. # 21). We have now referenced this paper more clearly in the revised manuscript and have also summarised this in the Methods section on page 6.

In addition to these specific changes we have made in response to the reviewer comments, on reviewing our analyses we note that, unfortunately, we found a minor error in the script underlying the multivariable analysis included in Table 3. We have now corrected that error and re-run the analysis which made no material difference to our initial conclusions. The revised p-values are now shown in the updated table. Finally, we have also made a number of small changes to deal with typographical errors and issues of clarity.